# Alternations of Lipoprotein Profiles in the Plasma as Biomarkers of Huntington’s Disease

**DOI:** 10.3390/cells12030385

**Published:** 2023-01-20

**Authors:** Kuo-Hsuan Chang, Mei-Ling Cheng, Chi-Jen Lo, Chun-Ming Fan, Yih-Ru Wu, Chiung-Mei Chen

**Affiliations:** 1Department of Neurology, Chang Gung Memorial Hospital Linkou Medical Center, Taoyuan 333, Taiwan; 2College of Medicine, Chang Gung University, Taoyuan 333, Taiwan; 3Department of Biomedical Sciences, Chang Gung University, Taoyuan 333, Taiwan; 4Metabolomics Core Laboratory, Healthy Aging Research Center, Chang Gung University, Taoyuan 333, Taiwan; 5Clinical Metabolomics Core Laboratory, Chang Gung Memorial Hospital, Taoyuan 333, Taiwan

**Keywords:** Huntington’s disease, biomarker, lipoprotein, high-density lipoprotein, low-density lipoprotein, very low-density lipoprotein

## Abstract

Alterations in lipid composition and disturbed lipoprotein metabolism are involved in the pathomechanism of Huntington’s disease (HD). Here, we measured 112 lipoprotein subfractions and components in the plasma of 20 normal controls, 24 symptomatic (sympHD) and 9 presymptomatic (preHD) HD patients. Significant changes were found in 30 lipoprotein subfractions and components in all HD patients. Plasma levels of total cholesterol (CH), apolipoprotein (Apo)B, ApoB-particle number (PN), and components of low-density lipoprotein (LDL) were lower in preHD and sympHD patients. Components of LDL4, LDL5, LDL6 and high-density lipoprotein (HDL)4 demonstrated lower levels in preHD and sympHD patients compared with controls. Components in LDL3 displayed lower levels in sympHD compared with the controls, whereas components in very low-density lipoprotein (VLDL)5 were higher in sympHD patients compared to the controls. The levels of components in HDL4 and VLDL5 demonstrated correlation with the scores of motor assessment, independence scale or functional capacity of Unified Huntington’s Disease Rating Scale. These findings indicate the potential of components of VLDL5, LDL3, LDL4, LDL5 and HDL4 to serve as the biomarkers for HD diagnosis and disease progression, and demonstrate substantial evidence of the involvement of lipids and apolipoproteins in HD pathogenesis.

## 1. Introduction

Huntington’s disease (HD) is a neurodegenerative disorder that is inherited in an autosomal dominant pattern, and is marked by a combination of involuntary chorea movements and various psychiatric symptoms [1]. HD is caused by an unstable expansion of a CAG trinucleotide repeat sequence in the huntingtin (HTT) gene, which encodes for a polyglutamine (polyQ) tract in exon 1 of the HTT protein [1]. This expansion causes the HTT protein to misfold and result in the formation of intranuclear and intracytoplasmic aggregates [1]. The accumulation of these toxic aggregates further lead to transcriptional dysregulation [2], mitochondrial and metabolic dysfunction [3], oxidative stress [3] and impaired proteasome activity [4], resulting in neuronal dysfunction and death [5]. These neurodegenerative mechanisms disturb cellular metabolism and produce specific protein and metabolite profiles [6,7,8], which can be detected not only in the central nervous system, but also in peripheral tissues [8,9,10]. Identifying peripheral biomarkers, particularly in blood, would assist in monitoring disease progression and therapeutic response in HD patients.

Lipoproteins are particles composed of triglyceride, cholesterol, phospholipids and apolipoproteins. By receptor-mediated endocytosis, lipoproteins transport their fat components from the extracellular fluid to cells. Circulatory lipoproteins are classified into seven groups based on their density, apolipoprotein content, and lipid composition [11]. High-density lipoproteins (HDLs) facilitate cholesterol efflux and exhibit anti-inflammatory, vasodilatory and endothelial protective properties [12], while low-density lipoproteins (LDL) participate in atherosclerosis [13,14]. Lipoproteins can also be sub-divided according to particle size into various subfractions. These lipoprotein subfractions may be associated with hypertriglyceridemia [15], diabetes mellitus [16] and atherosclerosis [13,14,17].

Previous studies have indicated an association between levels of components of lipoproteins in circulation and neurodegenerative diseases [18,19,20,21,22]. Elevated blood level of cholesterol in LDL were reported in patients with Alzheimer’s disease (AD) [18]. In contrast, cerebrospinal fluid levels of small HDL were positively correlated with the performance of cognitive function [19]. Similarly, higher levels of cholesterol in LDL and lower levels of cholesterol in HDL have been associated with a higher incidence of Parkinson’s disease (PD) [20,21,22]. However, the association of lipoprotein subfractions and components with HD has not yet been thoroughly explored. Therefore, the present study investigated alterations of circulatory lipoprotein profiles in HD patients using nuclear magnetic resonance (NMR) spectroscopy-based analysis. 

## 2. Materials and Methods

### 2.1. Ethics Statement and Study Participants

The study protocol was reviewed and approved by the Institutional Review Boards of Chang Gung Memorial Hospital (ethical license No: 202100892A3). All recruited patients and controls provided written informed consent.

### 2.2. Patient Recruitment and Plasma Preparation

Patients with HD and pre-symptomatic HD (preHD) were recruited from the neurology outpatient clinics of Chang Gung Memorial Hospital. The diagnosis was established through a neurological examination and genetic testing that revealed expanded CAG repeats in the exon 1 region of *HTT* [1]. Each patient’s symptoms were assessed using the Unified Huntington’s Disease Rating Scale (UHDRS) [23]. The scale ranges from normal to most severe, with total motor scores for motor assessment ranging from 0 to 124, independence scale from 100 to 10, and functional capacity from 13 to 0. The disease-burden score is determined using a formula (age × [CAG − 35.5]) [24]. The subject carrying a genetic mutation in the *HTT* without clinical symptoms with zero of the total motor score was defined as preHD. Normal controls (NC) were recruited from neurology outpatient clinics and were matched for sex and age. All participants were free from systemic infection, chronic renal failure, cardiac or liver dysfunction, malignancies, autoimmune diseases, stroke or other neurodegenerative diseases except HD. Patients and NC were instructed to avoid taking nutritional supplements, smoking, coffee and alcohol for at least one month. Blood samples were collected in EDTA-containing tubes after participants had fasted overnight for 12 h. The samples were centrifuged at 1500–2500× *g* for 15 min within one hour. The plasma (supernatant) was then transferred to a fresh 1.5-mL Eppendorf tube. All plasma samples were stored at −80 °C before measurements.

### 2.3. Nuclear Magnetic Resonance Analysis (^1^H NMR Spectroscopic Measurements)

A plasma sample (100 μL) was mixed with 75 mM pH 7.4 sodium phosphate in a 1:1 ratio, and 200 μL was transferred into a 3-mm Bruker SampleJet NMR tube. All NMR analysis was conducted using Bruker Avance III HD 600 MHz spectrometers, with a TXI probe. The sample cooling system of Bruker SampleJet set to 6 °C. For each sample, a solvent presaturation 1D 1H experiment (64 scans, 98,304 data points, spectral width of 18,028.85 Hz), followed by a 1D 1H Carr–Purcell–Meiboom–Gill (CPMG) spin-echo experiment (64 scans, 73,728 data points, spectral width of 12,019.23 Hz), was conducted. The data were acquired and automatically processed using Bruker Topspin 3.6.2 and ICON NMR (Bruker Biospin GmbH, Rheinstetten, Germany) to perform phasing, baseline correction, and calibration (TSP to 0 ppm). All equipment was provided by Bruker (Bruker Biospin GmbH, Rheinstetten, Germany).

The 112 lipoprotein subfractions were analyzed by the method from Bruker IVDr Lipoprotein Subclass Analysis (B.I.-LISA). The data were obtained by mathematically quantifying the peaks (−CH2, δ = 1.25 ppm and −CH3, δ = 0.8 ppm) of the 1D spectrum after normalization using the Bruker QuantRef manager within Topspin and a PLS-2 regression model [25]. The lipoprotein data include information on the chemical components of apolipoprotein-A1 (ApoA1), apolipoprotein-A2 (ApoA2), apolipoprotein-B (ApoB), apolipoprotein B100 (ApoB100), cholesterol (CH), free cholesterol (FC), phospholipid (PL), particle number (PN), triglyceride (TG), and the B100/A1 ratio in different classes of density, which are very low-density lipoprotein (VLDL, 0.950–1.006 kg/L), low-density lipoprotein (LDL, density 1.019–1.63 kg/L), intermediate-density lipoprotein (IDL, density 1.006–1.019 kg/L) and high-density lipoprotein (HDL, density 1.063–1.210 kg/L). The VLDL, LDL and HDL were further subdivided into specific density subfractions, with VLDL having 5 subfractions, LDL having 6 subfractions, and HDL having 4 subfractions [26].

### 2.4. Statistical Analysis

Continuous variables were described using mean and standard deviation (SD) and analyzed using Mann–Whitney U test with false discovery rate (FDR) adjustment, or Kruskal–Wallis with Dunn’s post hoc test when appropriate. Categorical variables, presented as count and percentages, were analyzed using Fisher’s exact test. Orthogonal partial least squares-discriminant-analysis (OPLS-DA) was used to analyze the relationship between clinical variables and metabolites. The score of variable importance in the projection (VIP) of each metabolite in the model was calculated to indicate its contribution to the classification. A higher VIP value indicates a stronger contribution to the discrimination between groups. A VIP value greater than 1.0 was considered to be significantly different. Pearson correlation was used to examine the association between the levels of metabolites and clinical parameters. Receiver operating characteristic (ROC) analysis was employed to evaluate the ability of individual molecules to differentiate HD patients from NC. Selected molecules were further analyzed using a support vector machine (SVM) algorithm. Model performance was evaluated by generating ROC curves through Monte-Carlo cross-validation using balanced sub-sampling. Two thirds of the subjects were used to build the classification models, which were then validated on the remaining one third. To create a smooth ROC curve, 100 cross-validations were conducted, and the results were averaged. All analyses were conducted using R software version 4.0.3 with the rstatix and metaboanalyst packages (R Fundation, Jaunpur, India).

## 3. Results

In this study, 33 genetic-confirmed HD patients, including 24 symptomatic (sympHD) and 9 presymptomatic (preHD) HD patients, and 20 NC were recruited (Table 1). Plasma concentrations of 122 lipoprotein subfractions were quantitated by NMR. The OPLS-DA of all metabolites was able to separate HD from NC (R2Y, 0.44, Q2, 0.30, Figure 1A), while 40 metabolites had VIP score greater than 1.0 (Figure 1B). Plasma levels of 29 lipoprotein subfractions or components were significantly lower in all HD patients compared with NC (Table 2). Lower levels of 26 lipoprotein subfractions or components were seen in sympHD patients, while 20 of them were also consistently lower in preHD patients compared to the NC. In contrast, the plasma level of VLDL5-FC was significantly higher in all HD and sympHD patients. The heatmap of hierarchical clustering using the selected biomarker candidates was shown in Figure 1C. Most of HD patients were aggregated in the same cluster. 

In sympHD patients, the HDL4-ApoA1 levels were inversely correlated with the total motor scores (r = −0.430, *p* = 0.036, Figure 2A), while the levels of HDL4-FC were positively correlated with the independence scores of UPDRS (r = 0.421, *p* = 0.041, Figure 2B). The levels of LDL3-FC (r = 0.437, *p* = 0.033, Figure 2C) and LDL3-PL (r = 0.413, *p* = 0.045, Figure 2D) were positively correlated with the independence scores of UPDRS. The levels of LDL6-FC were positively correlated with the ages at onset of sympHD patients (r = 0.419 *p* = 0.042, Figure 2E). The levels of VLDL5-FC were positively correlated with the total motor scores (r = 0.545, *p* = 0.006, Figure 2F), and inversely correlated with the independence scores (r = −0.624, *p* = 0.001, Figure 2G) and functional capacity (r = −0.511, *p* = 0.012, Figure 2H) of UPDRS. These evident correlations suggest that lipoprotein subfractions may serve as potential biomarkers to indicate disease progression and disability in HD.

Analysis of the ROC curve was conducted to assess the diagnostic potential of 30 selected lipoprotein components as biomarkers for HD (Figure 3). The analysis revealed that LDL5-FC and LDL4-FC had the highest area under the ROC curve (AUC) for distinguishing HD from normal controls (AUC, 0.891, Figure 3A,B), followed by LDL-CH (0.864, Figure 3C) and LDL-PL (0.863, Figure 3D). The SVM algorithm using LDL5-FC, LDL4-FC, LDL-CH and LDL-PL demonstrated a good ability to differentiate HD from NC (AUC: 0.902, Figure 3E). These findings support the prospective of developing a machine-learning algorithm using the data of lipoprotein subfractions for diagnosis of HD.

## 4. Discussion

By comprehensively examining the plasma levels of 122 lipoprotein fractions and components in HD patients, we found profound alterations in levels of 30 lipoprotein fractions or components in preHD and sympHD patients. Levels of the components of lipoprotein subfractions HDL3, HDL4, LDL3, LDL4 and LDL5 were reduced in HD patients. HDL4-ApoA1, HDL4-CH, LDL3-FC and LDL3-PL demonstrated negative correlations with the functional disabilities of HD patients. Contrarily, levels of VLDL5-FC were elevated and positively correlated with the functional disabilities in HD patients. LDL5-FC, LDL4-FC, LDL-CH and LDL-PL further displayed potential to identify patients with HD. To our knowledge, this is the first study to use NMR spectrometry to explore lipoprotein profiles in HD. These comprehensive compositional analyses of lipoproteins identify components in lipoprotein subfractions that could serve as novel biomarkers as well as potential pharmacological targets for HD. 

Our study found the levels of HDL3-FC, HDL4-CH, HDL4-ApoA1 and HDL4-FC to be significantly decreased in HD patients. It has been shown that HDL may demonstrate anti-apoptotic, anti-oxidant, anti-thrombotic and anti-inflammatory activities [27]. Plasma HDL-CH levels are significantly correlated with cognitive function in elderly populations [28,29,30,31]. Lower levels of plasma HDL-CH have been linked to a higher risk of developing AD and PD [19,21]. Low plasma levels of HDL-CH are also associated with depression [32,33]. The delivery of modified HDLs across blood brain barriers has been shown to reduce the accumulation of Aβ, decrease microglia activation, and alleviate neurological damage, as well as rescue memory deficits in a mouse model of AD [34]. Previous studies have found reduced levels of Apo-A1 in the CSF or in the plasma of PD patients [35,36,37]. Plasma levels of Apo-A1 were positively correlated with the degree of dopamine transporter uptake. Further studies will be needed to investigate the mechanisms of how HDL3 and HDL4 involved in HD pathogenesis. 

Taken up by the endothelium and macrophage, LDLs under oxidation stimulate atherogenic plaque formation. Plasma levels of LDL-CH are elevated in AD and PD [18,20,22]. However, the Honolulu-Asia Aging Study found correlation between low levels of LDL-CH and an increased risk of PD in men aged 71–75 years [38]. In HD patients, our study found reduced levels not only in LDL-CH, but also in components of small dense fractions of LDL, such as LDL3, LDL4 and LDL5. Our results, for the first time, demonstrate the potential role of small dense LDLs in HD. We further found negative correlations between LDL3-PL or LDL3-FC and the independence scores of UHDRS. Compared with large dense LDLs, small dense LDLs are more likely to cause atherosclerosis because they are more easily able to penetrate the artery wall and have a longer presence in the bloodstream, making them more susceptible to oxidation [39]. Elevation of LDL5-CH and LDL6-CH has also been identified in patients with cerebral small vascular disease [40]. Further research is needed to better understand the role of small, dense LDLs in the pathogenesis of HD and their impact on functional abilities.

VLDL carries most of the TG in the blood. Elevated levels of VLDL-CH have been linked to increased carotid intima-media thickness and the presence of atherosclerotic plaques [41,42]. Usually, VLDLs cannot penetrate the blood–brain barrier (BBB). However, metabolic stress could weaken the BBB and allow VLDL to enter the central nervous system [43,44]. Tail vein injection of electronegative VLDLs to mice can trigger microglial activation and cognitive dysfunction [44]. The uptake of VLDL by microglia is associated with lipoprotein lipase (LPL) dysfunction [45], which affects cognitive function and increases risks of AD [45,46]. Here we found elevated VLDL5-FC in plasma of HD patients. The correlations of VLDL5-FC levels and the scores in UPDRS motor assessment, independence and function capacity may further indicate its potential role in pathogenesis of HD, whereas the underlying mechanism needs to be investigated by further studies. 

Under physiological conditions, ATP-binding cassette transporter A1 (ABCA1) transports phospholipid and cholesterol in cells to extracellular pre-β HDL to form HDL4 [47]. Lipoprotein lipase hydrolyzes TGs or free fatty acids in VLDL particles to form LDL [17]. Free cholesterols in small HDLs (HDL3 and HDL4) are converted to cholesterol esters (CEs) by lecithin cholesterol acyl transferase. Cholesteryl ester transfer protein (CETP) then facilitates the transfer of CEs from HDLs to LDLs in exchange for TGs [48]. In HD, mutant HTT down-regulates the expression of ABCA1, thus affecting the transport of cholesterol to form HDL4 [49]. Hepatocyte-specific *ABCA1* knock-out mice demonstrate impaired HDL production accompanied with increased amount of VLDL and enhanced LDL catabolism [49]. These findings are consistent to our results (Figure 4), which showed up-regulation of VLDL5 components and down-regulation of components of small dense LDLs (LDL3, LDL4, LDL5) and HDLs. Low serum levels of ABCA1 have been observed in AD patients [50]. Knockout of ABCA1 in *APP* transgenic mouse model for AD lead to impaired learning and memory retention [51]. Further investigation will be warranted to elucidate the regulatory network between ABCA1, LPL and CETP, and how lipoproteins participate in the pathogenesis of neurodegeneration in HD. 

This study has several limitations. First, it may be underpowered to detect subtle changes in lipoprotein subfractions in HD. Second, the small proportion of patients with preHD may limit the detection of lipoprotein subfraction alterations in this group of patients. Finally, the potential unknown interactions of medications may also play a role in the metabolic differences between groups. Nevertheless, our study, for the first time, presents alterations of lipoprotein components in plasma of HD patients. Future investigations using large and independent cohorts are required to further validate our findings. 

## Figures and Tables

**Figure 1 cells-12-00385-f001:**
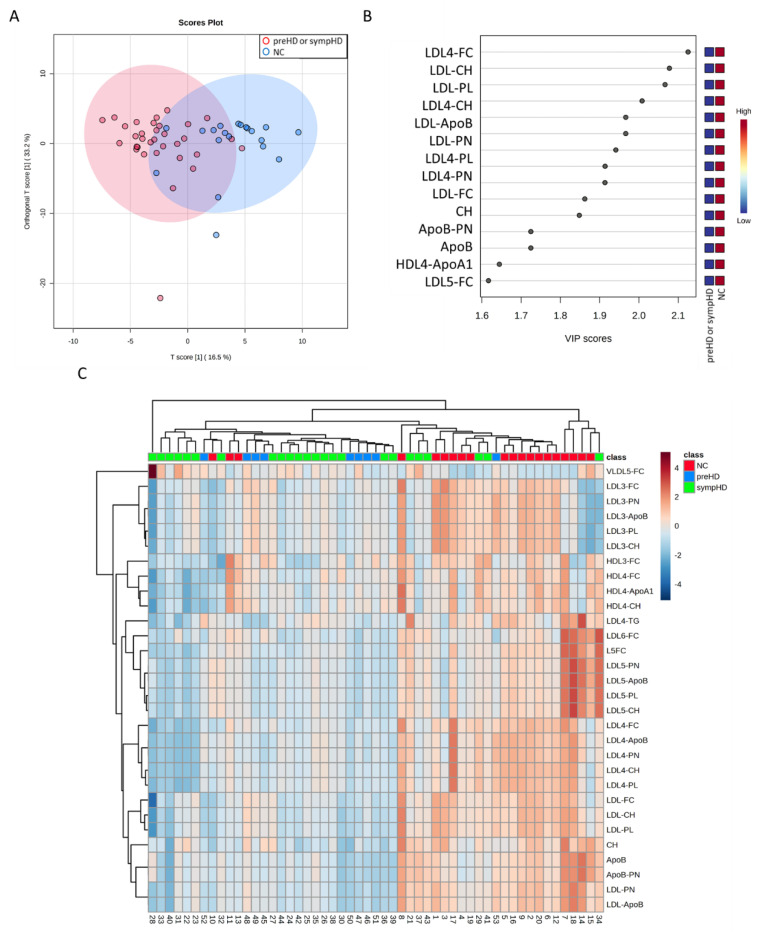
Orthogonal partial least squares-discriminant analysis (OPLS-DA) of normal controls (NC, n = 20) and patients with symptomatic Huntington’s disease (sympHD) (n = 24) or pre-symptomatic HD (preHD, n = 9). (**A**) OPLS-DA of metabolites shows a separation between the two groups (R^2^Y = 0.44, Q^2^ = 0.30). R^2^Y, cumulative variation in the Y matrix; Q^2^, predictive performance of the model. (**B**) Lipoprotein components with a variable importance in projection (VIP) score greater than 1.6, indicating their contribution to the OPLS-DA model. (**C**) Heatmap of the hierarchical clustering. The dendrogram at the top shows the clustering of patients, and the dendrogram on the side shows the clustering of metabolites. The colors at the top of the heatmap represent NC, HD or preHD. The colors in the heatmap represent normalized intensities, scaled to a mean of zero and unit variance for each metabolites. ApoA1, apolipoprotein A1; ApoB, apolipoprotein B; CH, cholesterol; FC, free cholesterol; HDL, high-density lipoprotein; LDL, low-density lipoprotein; PL, phospholipid; PN, particle number; VLDL, very low-density lipoprotein.

**Figure 2 cells-12-00385-f002:**
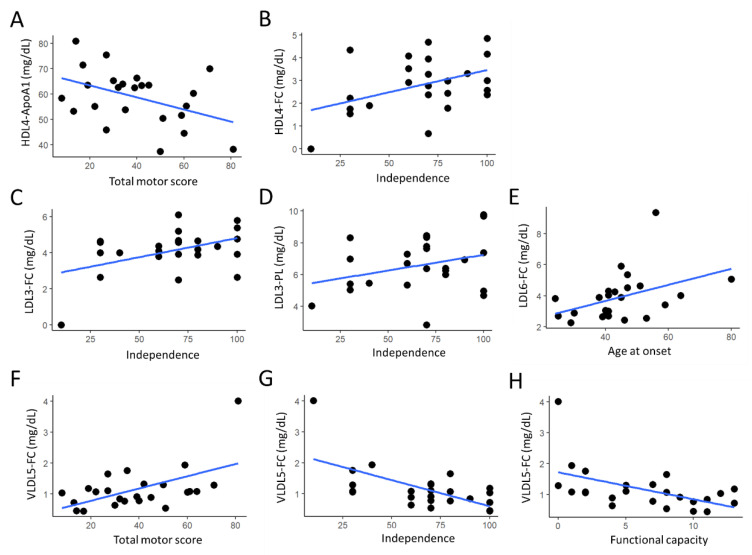
(**A**–**H**) The correlation between identified lipoprotein components and clinical parameters (Unified Huntington’s Disease Rating Scale including total motor score, independence score and functional capacity; age at onset) in symptomatic Huntington’s disease patients. Dots represent the data points for each patient. Blue lines denote the regression lines. ApoA1, apolipoprotein A1; FC, free cholesterol; HDL, high-density lipoprotein; LDL, low-density lipoprotein; PL, phospholipid; VLDL, very low-density lipoprotein.

**Figure 3 cells-12-00385-f003:**
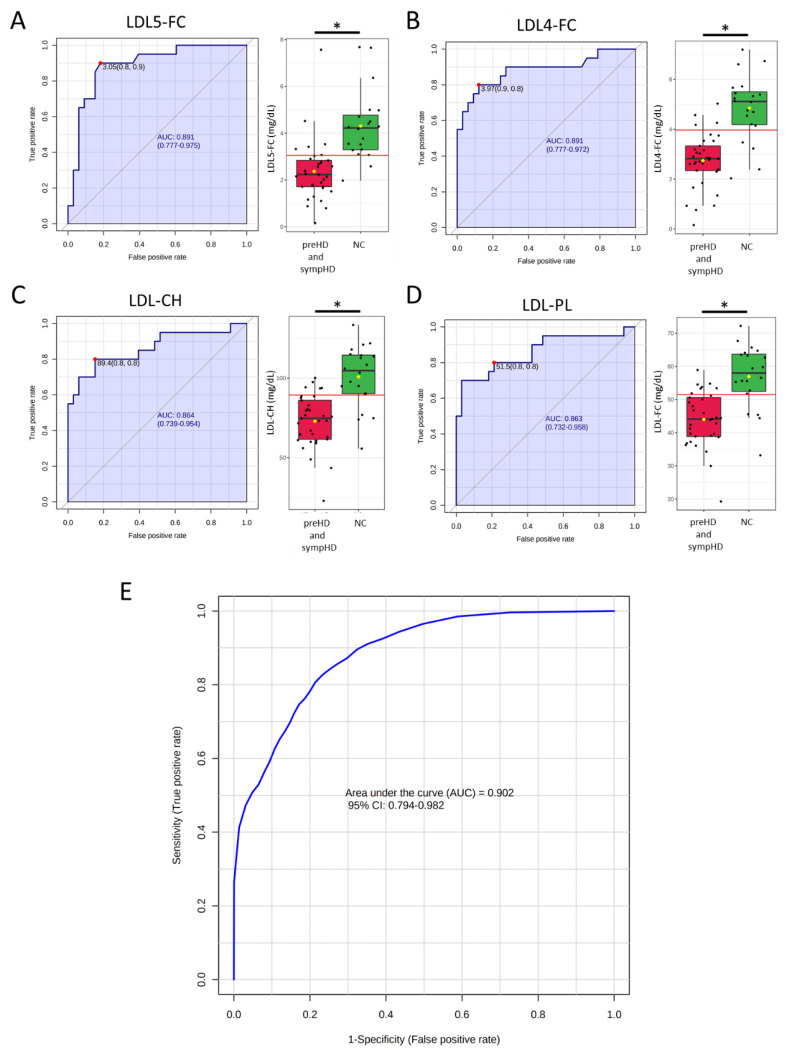
The receiver operating characteristic (ROC) curve and the box plot for plasma levels of (**A**) LDL5-FC, (**B**) LDL4-FC, (**C**) LDL-CH and (**D**) LDL-PL. Area under the ROC curve (AUC) was shaded. The black center line in the box plot indicated the median, while the red or green boxes represented the 25th to 75th percentiles. The black whiskers marked the 5th and 95th percentiles, and mean values were represented by yellow diamonds. The red dots and lines denoted the optimal cut-off. *: Statistically significant between two groups, *p* < 0.05, Mann–Whitney U test. (**E**) ROC analysis of the above four lipoprotein components using support vector machine. CH, total cholesterol; FC, free cholesterol; LDL, low-density lipoprotein.

**Figure 4 cells-12-00385-f004:**
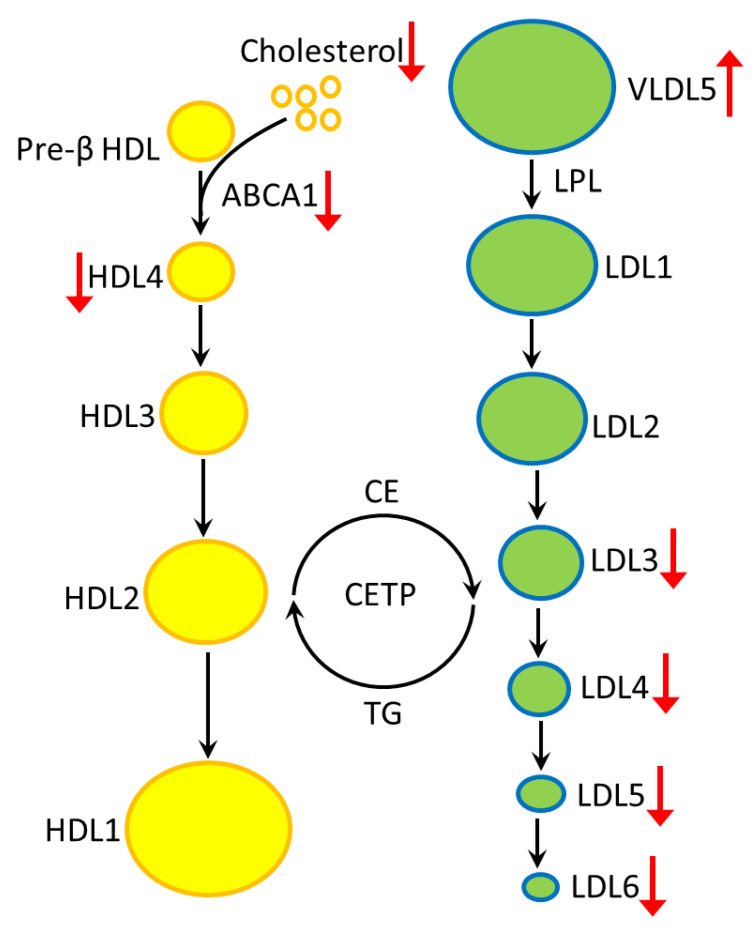
The altered profiles of lipoproteins in plasma of HD are summarized. HDL4, LDL3, LDL4 and LDL5 are reduced, while VLDL5 is increased in the plasma of HD patients. The reduced HDL4 may be generated by reduced ABCA1-mediated cholesterol efflux or decreased cholesterol generation. The mechanism of how ABCA1 deficiency affects VLDL and LDL subfractions remains uncertain. ABCA1, ATP-binding cassette transporter A1; CE, cholesteryl ester; CETP, cholesteryl ester transfer protein; HDL, high-density lipoprotein; LDL, low-density lipoprotein; LPL, lipoprotein lipase; TG, triglyceride; VLDL, very low-density lipoprotein.

**Table 1 cells-12-00385-t001:** Demographic characteristics and blood biochemical parameters of included the patients with Huntington’s disease (HD) and the normal controls (NC).

	NC	HD		
	(n = 20)	preHD (n = 9)	sympHD (n = 24)	All (n = 53)
Age (years)	52.70 ± 8.95	32.00 ± 7.93 *	50.00 ± 10.81	47.96 ± 12.14
Male (%)	10 (50.00)	1 (11.11) **	16 (66.66)	427(50.94)
BMI	23.42 ± 2.57	20.24 ± 2.41	21.87 ± 2.54	22.16 ± 2.77
Pre-prandial glucose (mg/dL)	93.30 ± 10.74	87.80 ± 7.93	98.34 ± 16.59	93.38 ± 12.15
UHDRS				
Total motor score		0	39.21 ± 19.30	
Independence scale		100	66.67 ± 25.93	
Functional capacity		13	6.38 ± 4.16	
Disease burden		272.56 ± 182.39	457.50 ± 124.61	
Medications				
Tetrabenazine (%)	0	0	6 (25.00)	
Benzodiazepines (%)	3 (15.00)	0	14 (58.33)	
Antipsychotics (%)	0	0	11 (45.83)	
Antidepressants (%)	0	0	8 (33.33)	
Ubidecarenone (%)	0	0	10 (41.67)	

BMI’ body mass index; HD’ Huntington’s disease, preHD’ presympatomatic Huntington’s disease; sympHD’ symptomatic Huntington’s disease; UHDRS’ Unified Huntington’s Disease Rating Scale. *: Statistically significant in comparison with NC and sympHD, *p* < 0.05. Kruskal–Wallis test with Dunn’s correction. **: Statistically significant in comparison with NC and sympHD, *p* < 0.05. Fisher’s exact test.

**Table 2 cells-12-00385-t002:** Significant alterations in levels of plasma lipoprotein components in the patients with Huntington’s diseases (HD) compared to the normal controls (NC).

Metabolite Name	NC	HD		
	(n = 20)	preHD (n = 9)	sympHD (n = 24)	All (n = 33)
ApoB (mg/dL)	77.39 ± 13.06	58.96 ± 8.63 **	63.32 ± 11.52 **	62.13 ± 10.98 *
ApoB-PN (nmol/L)	1407.08 ± 237.50	1072.05 ± 157.06 **	1151.33 ± 209.48 **	1129.71 ± 199.72 *
CH (mg/dL)	178.49 ± 25.35	152.77 ± 19.12 **	155.05 ± 21.88 **	154.43 ± 21.19 *
LDL-ApoB (mg/dL)	61.00 ± 10.23	48.32 ± 9.09 **	49.83 ± 9.54 **	49.42 ± 9.44 *
LDL-CH (mg/dL)	100.83 ± 19.08	78.80 ± 16.14 **	71.86 ± 16.64 **	72.94 ± 16.60 *
LDL-FC (mg/dL)	30.67 ± 5.68	24.15 ± 4.90 **	22.83 ± 5.70 **	23.19 ± 5.53 *
LDL-PL (mg/dL)	56.96 ± 9.23	44.93 ± 7.92 **	43.64 ± 8.28 **	43.99 ± 8.20 *
LDL-PN (nmol/L)	1163.74 ± 186.01	878.58 ± 165.33 **	906.04 ± 173.41 **	898.55 ± 171.68 *
LDL3-ApoB (mg/dL)	9.34 ± 3.33	8.10 ± 2.07	6.54 ± 2.19 **	6.96 ± 2.26 *
LDL3-CH (mg/dL)	16.02 ± 6.96	13.92 ± 4.05	10.82 ± 3.94 **	11.67 ± 4.21 *
LDL3-FC (mg/dL)	5.76 ± 2.02	4.97 ± 1.22	4.11 ± 1.22 **	4.34 ± 1.28 *
LDL3-PL (mg/dL)	9.05 ± 3.40	7.93 ± 1.99	6.42 ± 2.08 **	6.83 ± 2.16 *
LDL3-PN (nmol/L)	169.85 ± 60.53	147.23 ± 37.73	118.89 ± 39.74 **	126.62 ± 41.18 *
LDL4-ApoB (mg/dL)	8.50 ± 2.70	4.68 ± 2.26 **	4.32 ± 2.72 **	4.41 ± 2.61 *
LDL4-CH (mg/dL)	14.29 ± 5.03	7.76 ± 3.25 **	6.89 ± 4.36 **	7.13 ± 4.11 *
LDL4-FC (mg/dL)	4.85 ± 1.31	3.00 ± 0.72 **	2.64 ± 1.19 **	2.74 ± 1.10 *
LDL4-PL (mg/dL)	8.04 ± 2.55	4.76 ± 1.67 **	4.36 ± 2.45 **	4.47 ± 2.27 *
LDL4-PN (nmol/L)	154.61 ± 49.05	85.00 ± 41.15 **	78.47 ± 49.51 **	80.25 ± 47.47 *
LDL4-TG (mg/dL)	1.85 ± 0.74	1.11 ± 0.68	1.29 ± 0.67	1.24 ± 0.67 *
LDL5-ApoB (mg/dL)	9.46 ± 4.68	3.85 ± 1.41 **	5.39 ± 3.76 **	4.97 ± 3.36 *
LDL5-CH (mg/dL)	13.88 ± 6.51	5.37 ± 2.16 **	7.33 ± 5.50 **	6.80 ± 4.90 *
LDL5-FC (mg/dL)	4.31 ± 1.46	2.12 ± 0.51 **	2.45 ± 1.47 **	2.36 ± 1.29 *
LDL5-PL (mg/dL)	7.73 ± 3.28	3.41 ± 1.08 **	4.42 ± 2.83 **	4.14 ± 2.52 *
LDL5-PN (nmol/L)	172.02 ± 85.22	70.04 ± 25.68 **	97.98 ± 68.34 **	90.36 ± 61.08 *
LDL6-FC (mg/dL)	5.25 ± 1.65	3.19 ± 0.91 **	3.90 ± 1.50 **	3.70 ± 1.40 *
HDL3-FC (mg/dL)	2.04 ± 0.59	1.76 ± 0.38	1.54 ± 0.55	1.60 ± 0.52 *
HDL4-ApoA1 (mg/dL)	70.93 ± 11.06	60.42 ± 5.64 **	58.92 ± 10.66 **	59.33 ± 9.58 *
HDL4-CH (mg/dL)	18.51 ± 3.92	15.76 ± 2.95	14.90 ± 4.01	15.13 ± 3.77 *
HDL4-FC (mg/dL)	3.87 ± 1.06	2.99 ± 0.60	2.81 ± 1.19	2.86 ± 1.06 *
VLDL5-FC (mg/dL)	0.60 ± 0.39	0.90 ± 0.35	1.16 ± 0.71 **	1.09 ± 0.64 *

preHD, presympatomatic Huntington’s disease; sympHD, symptomatic Huntington’s disease. *: Statistically significant in comparison with NC. *p* < 0.05, Mann–Whitney U test with false discovery rate adjustment. **: Statistically significant in comparison with NC. *p* < 0.05, Kruskal–Wallis with Dunn’s post-hoc test.

## Data Availability

The datasets of the current study are available from the corresponding author on reasonable request.

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
