# Peer review of "Alternations of Lipoprotein Profiles in the Plasma as Biomarkers of Huntington’s Disease"

_cells, 2023, doi:10.3390/cells12030385_

Round 1

Reviewer 1 Report

The study by Chang and colleagues evaluates the levels of several lipoproteins in plsma from human HD patients, including a small cohort of presymptomaic individuals. 

The manuscript is concise and provides interesting information. It is well written and the figures are well presented. 

Minor point:

- It would be interesting to see the heatmap clustering (or at least adding the color to differnciate in the dendrogram) by control, presymptomatic and HD.

Author Response

We appreciate all reviewers giving their constructive comments that helped us to improve the paper. Please find attached point by point reply to reviewers. We also extensively revised the text to reduce duplication. The revised sections are highlighted in yellow colour.

Reviewer 1:

The study by Chang and colleagues evaluates the levels of several lipoproteins in plsma from human HD patients, including a small cohort of presymptomaic individuals. 

The manuscript is concise and provides interesting information. It is well written and the figures are well presented. 

Minor point:

- It would be interesting to see the heatmap clustering (or at least adding the color to differnciate in the dendrogram) by control, presymptomatic and HD.

Response: Thank you for the suggestion. We have added the different color to differentiate control, presymptomatic HD and HD in the dendrogram.

Sincerely yours,

Chiung-Mei Chen, M.D., Ph.D.

Reviewer 2 Report

The authors studied the lipoprotein profile of a relatively large cohort of HD and preHD subjects. The study is well written and clear. The results suggest that lipoprotein subfraction alteration could be associated with HD pathogenesis and that lipoprotein profile can differentiate HD from controls. This is an interesting exploratory study that can add knowledge in the field. 

I have only minor issues:

It is not clear what are the criteria to define PreHD (i.e. TMS cut-off). Please add in the methods

If available, add UHDRS evaluations also for PreHD subjects

Author Response

We appreciate all reviewers giving their constructive comments that helped us to improve the paper. Please find attached point by point reply to reviewers. We also extensively revised the text to reduce duplication. The revised sections are highlighted in yellow colour.

Reviewer 2:

The authors studied the lipoprotein profile of a relatively large cohort of HD and preHD subjects. The study is well written and clear. The results suggest that lipoprotein subfraction alteration could be associated with HD pathogenesis and that lipoprotein profile can differentiate HD from controls. This is an interesting exploratory study that can add knowledge in the field. 

I have only minor issues:

It is not clear what are the criteria to define PreHD (i.e. TMS cut-off). Please add in the methods

 Response:

We have added a sentence to define preHD:The subject carrying a genetic mutation in the HTT without clinical symptoms with zero of the total motor score was defined as preHD.” in section of Patient recruitment and plasma preparation.

If available, add UHDRS evaluations also for PreHD subjects

Response:

We also added the UHDRS evaluations for preHD subjects in Table 1.

Sincerely yours,

Chiung-Mei Chen, M.D., Ph.D.